# SNP rs6564851 in the BCO1 Gene Is Associated with Varying Provitamin a Plasma Concentrations but Not with Retinol Concentrations among Adolescents from Rural Ghana

**DOI:** 10.3390/nu12061786

**Published:** 2020-06-16

**Authors:** Sophie Graßmann, Olga Pivovarova-Ramich, Andrea Henze, Jens Raila, Yaw Ampem Amoako, Richard King Nyamekye, George Bedu-Addo, Frank P. Mockenhaupt, Matthias B. Schulze, Ina Danquah

**Affiliations:** 1Department Molecular Epidemiology, German Institute of Human Nutrition Potsdam-Rehbruecke (DIfE), 14558 Nuthetal, Germany; s.grassmann@yahoo.de (S.G.); mschulze@dife.de (M.B.S.); 2Research Group Molecular Nutritional Medicine, Department of Molecular Toxicology, German Institute of Human Nutrition Potsdam-Rehbruecke (DIfE), 14558 Nuthetal, Germany; olga.ramich@dife.de; 3Department of Endocrinology, Diabetes and Nutrition, Charité—Universitaetsmedizin Berlin, 13125 Berlin, Germany; 4German Center for Diabetes Research (DZD), 85764 Munich-Neuherberg, Germany; 5Junior Research Group ProAID, Institute of Nutritional Science, Potsdam University, 14558 Nuthetal, Germany; henze@uni-potsdam.de; 6Physiology and Pathophysiology of Nutrition, Institute of Nutrition Science, Potsdam University, 14558 Nuthetal, Germany; jens.raila@uni-potsdam.de; 7Komfo Anokye Teaching Hospital, Kwame Nkrumah University of Science and Technology (KNUST), P.O. Box 1934 Kumasi, Ghana; yamoako2002@yahoo.co.uk (Y.A.A.); gbeduaddo@gmail.com (G.B.-A.); 8Agogo Presbyterian Hospital, P.O. Box 27 Agogo, Ghana; kwajes@yahoo.com; 9Institute of Tropical Medicine and International Health, Charité—Universitaetsmedizin Berlin, 13353 Berlin, Germany; frank.mockenhaupt@charite.de; 10Heidelberg Institute of Global Health (HIGH), Universitaetsklinikum Heidelberg, 69120 Heidelberg, Germany

**Keywords:** carotenoids, vitamin A deficiency, single nucleotide polymorphism, BCO1, Ghana

## Abstract

In sub-Saharan Africa, vitamin A deficiency constitutes a severe health problem despite various supplementation and food fortification programs. Given that the intake of preformed vitamin A from animal products remains low in these countries, an efficient metabolization of plant-based provitamin A carotenoids is essential. Previously, adolescents in rural Ghana have shown high total plasma carotenoid concentrations, while 36% had a vitamin A deficiency (defined as plasma retinol < 0.7 µmol/L). Hence, the aim of this cross-sectional study was to identify the relationships between variants in the β-carotene 15,15’-oxygenase (BCO1) gene and plasma carotenoid concentrations among 189 15-year-old girls and boys in rural Ghana. *BCO1* rs6564851, rs7500996, rs10048138 and *PKD1L2* rs6420424, and rs8044334 were typed, and carotenoid concentrations were compared among the different genotypes. G allele carriers of rs6564851 (53%) showed higher plasma carotenoid concentrations than T allele carriers (median (interquartile range): 3.07 (2.17–4.02) vs. 2.59 (2.21–3.50) µmol/L, *p*-value = 0.0424). This was not explained by differences in socio-demographic or dietary factors. In contrast, no differences in plasma retinol concentrations were observed between these genotypes. Pending verification in independent populations, the low conversion efficiency of provitamin A carotenoids among rs6564851 G allele carriers may undermine existing fortification and supplementation programs to improve the vitamin A status in sub-Saharan Africa.

## 1. Introduction

Malnutrition and micronutrient deficiencies contribute to major health problems in developing countries. Looking at vitamin A, an insufficient sustenance is observed in about 250 million pre-school children [1] and 19.1 million pregnant women. Plasma retinol concentrations < 0.7 µmol/L define vitamin A deficiency (VAD), with sub-Saharan Africa and South East Asia showing the highest prevalence [2,3]. The main symptoms of VAD comprise visual impairment (xerophthalmia and night blindness), and increased morbidity and mortality from infectious diseases (e.g., measles and diarrhea) caused by an impaired immune system [4]. During pregnancy, low plasma retinol concentrations have a negative impact on cell differentiation and proliferation, thereby leading to disturbed embryonal development and growth retardation of the child [5,6].

Vitamin A can either be consumed as preformed vitamin A (retinol, retinyl palmitate) from animal-derived products (meat, especially liver, and dairy products) or as carotenoids with provitamin A (proVA) activity (α-carotene, β-carotene, β-cryptoxanthin) from plant-based foods (yellow- and orange-fleshy fruits and vegetables, green leaves) [3,7]. However, African diets are low in the consumption of animal products and proVA mainly contributes to the vitamin A supply [8]. In the enterocytes, 55–75% of the absorbed carotenoids are cleaved centrally into one (α-carotene, β-cryptoxanthin) or two (β-carotene) molecules of (all-*trans*-) retinal by the cytoplasmatic protein β-carotene 15,15’-oxygenase (BCO1), which is the key cleaving enzyme in the vitamin A metabolism. BCO1 is mainly expressed in the small intestine, but also in other organs like the liver, kidney, reproductive tissues, skin, and eyes, indicating a need of vitamin A in these tissues [9,10]. The synthesized retinal can further be metabolized to (all-*trans*-) retinol if required, or to transcriptionally active retinoic acid [11]. To avoid the accumulation of toxic amounts of retinol in the body, the cleavage of proVA is regulated by a negative feedback loop. Synthesized retinoic acid binds to retinoic acid receptors (RARs), which induces the expression of intestinal transcription factor intestine specific homeobox (ISX). The induction of ISX inhibits the expression of scavenger receptor class B member 1 (SR-BI) and BCO1, and hence leads to a reduction in the uptake and cleavage of dietary carotenoids [12].

Alicke et al. previously analyzed data of 188 adolescents from the Agogo 2000 birth cohort study to characterize “the co-occurrence of infectious diseases, malnutrition and cardio-metabolic risk factors” in rural Ghana. In this population, more than one-third (36%) had VAD, and the median plasma retinol concentration was 0.77 (IQR: 0.49–1.05) µmol/L [13]. However, the total plasma carotenoid concentrations (median: 2.8 (2.2–3.8) µmol/L) and β-carotene concentrations (1.49 (1.12–2.22) µmol/L) were three times higher than the reference values [14,15]. This was also seen among Nigerian mother–child pairs, who showed lower retinol and higher plasma carotenoid concentrations than mother–child pairs in the United States [16].

Many genetic variants in enzymes associated with the vitamin A metabolism were identified and associated with altered functionality [17,18,19,20,21]. As the conversion of β-carotene is mediated by BCO1, we focused on five genetic variants in this enzyme to explain the observed low retinol but high carotenoid concentrations in the adolescents in the Agogo 2000 birth cohort study. The BCO1 variant rs6564851 is the best described candidate in the literature, showing higher β-carotene and α-carotene concentrations in carriers of the G allele, caused by a decreased conversion efficiency of the BCO1 enzyme [17,18,21,22]. Further, the single nucleotide polymorphisms (SNPs) rs7500996 and rs10048138 have an impact on lutein and zeaxanthin concentrations [18,21]. The A allele of SNP rs10048138 is associated with higher concentrations of these carotenoids [18]. SNPs rs6420424 and rs8044334 have been reported by Ferrucci et al. and Lietz et al., with carriers of the A and G allele showing higher plasma β-carotene concentrations, due to a decrease of the BCO1 conversion efficiency [17,22]. Thus, genetic variants in the proVA cleaving enzyme can influence the vitamin A production in the human body.

Food fortification and vitamin A supplementation have reduced VAD prevalence in many affected countries, with the exception of sub-Saharan Africa [2]. The reasons for this still remain to be uncovered. Potential determinants comprise food-related (matrix, composition) and host-related factors (socio-demographic factors, health and nutritional status, genetics). While ethnic background has already been coined as a predictor of carotenoid and retinol concentrations, the underlying mechanisms are poorly understood [23,24].

Owing to the previously observed conflicts between retinol and carotenoid concentrations, and due to the lack of data on BCO1-relevance in this study population, the aim of this secondary, cross-sectional analysis was to identify the relationships between relevant genetic variants in the BCO1 gene and plasma carotenoid concentrations among Ghanaian adolescents in the Agogo 2000 birth cohort study. The primary aim of the study was to establish the impact of gestational malaria on health outcomes in later life.

## 2. Materials and Methods

### 2.1. Study Design and Population

The study was conducted in Agogo, which is located in the forested hills of the Ashanti Akim North District, central Ghana. The main income sources of the 170,000 inhabitants in this region are farming, trading and mining. Among pre-school children in Ghana, severe VAD prevalence (≥ 20%) is a public health problem [25].

Between June and August 2015, 200 adolescent boys and girls from the Agogo 2000 birth cohort were reassessed and underwent a health check-up as part of the follow-up on the impact of gestational malaria on health outcomes [26]. The inclusion criteria for this cross-sectional study were turning 15 in the year that the study was conducted, informed written consent, absence of pregnancy and no previous diagnosis of type 1 diabetes [13]. After excluding participants with implausible values (*n* = 1), without plasma carotenoid measurements (*n* = 1), and one of each twin-pair (*n* = 9) to avoid bias in the gene frequency, the final number of 189 adolescents were included in the analyses.

The study protocol was reviewed and approved by the Ethics Committee of the Kwame Nkrumah University of Science and Technology, Kumasi (CHRPE/AP/446/18). Written and informed consent was obtained from all caregivers and assent was given by all participants.

### 2.2. Physical Examination

Body weight was taken in light clothes and to the nearest 0.5 kg (Camry Person Scale, Model DT602, Hong Kong, China) and height was measured to the nearest 0.1 cm (Seca 213, Hamburg, Germany). Body mass index (BMI) was calculated by dividing a person’s weight (kg) by his/her squared height (m). Further, z-scores were calculated to compare the BMI and height of the Ghanaian adolescents with the median value of a healthy reference population (according to the World Health Organization (WHO)) of the same age and sex, using the soft-ware package AnthroPlus (version 1.0.4, WHO, Geneva, Switzerland). The calculated BMI-for-age z-scores (BAZ) and height-for-age z-scores (HAZ) were categorized according to the WHO. BAZ ≥ 2 was considered as obese, 1 ≤ BAZ < 2 as overweight, and BAZ < −2 as underweight in adolescents. HAZ < −2 was defined as stunting. Axillary body temperature was measured.

### 2.3. Questionnaire-Based Interviews

Demographic (age, sex, ethnic group, residence, place of school) and socioeconomic data (literacy status of the child, parental education and occupation, number of people living in the household, number of siblings) were documented in questionnaire-based interviews by trained personnel. The number of existing household assets (electricity, pipe-borne water, radio, TV, fan, cupboard, fridge, bicycle, motorbike, car, cattle) was recorded, and the proportion of 11 possible assets was calculated and presented as the wealth score. Also, medical history was documented.

The usual weekly intake of 10 food groups (including 80 food items) was documented using a semi-quantitative food frequency questionnaire (FFQ). The West African Food Composition Table (2012) was used to calculate the daily energy intake as well as the daily intakes of macronutrients (carbohydrates, fat, protein) in g/d and relevant micronutrients (dietary fiber, retinol, retinol activity equivalents (RAE)) in µg/d. Common Ghanaian household measures were used to estimate the portion sizes. Furthermore, the interviewers assessed the frequency and the duration of physical activity at school and in the spare time. From that, physical activity in minutes per day was calculated.

### 2.4. Identification of Releveant BCO1 SNPs

Literature was searched for variants in the BCO1 gene with a documented impact on plasma carotenoid concentrations in humans. Using Ensembl (version 95 [27]), minor allele frequencies and SNPs in linkage disequilibrium (LD) of 16 variants were identified. Due to the small study population, SNPs with an allele frequency < 30% in African reference populations (Ensembl version 95 [27]) (*n* = 8) and variants in LD (D’ ≥ 0.8; r² > 0.5) (*n* = 2) were excluded, yielding six BCO1 gene variants selected for genotyping.

### 2.5. Laboratory Analyses

Fasting venous blood samples were drawn into ethylenediamine tetra-acetic acid (EDTA) to measure the biomarkers of vitamin A metabolism and undernutrition, and into DNA-stabilizing buffer. Laboratory analyses were performed protected from sunlight exposure and within 4 h after venous blood collection. Plasma was separated by centrifugation at 8000 rpm for 10 min, and immediately stored at −80 °C. Full blood and plasma aliquots were transported to Germany on dry ice and stored at −80 °C.

#### 2.5.1. Biomarkers of Vitamin A Metabolism and Undernutrition

Plasma samples were analyzed in the Department of Physiology and Pathophysiology of Nutrition (University of Potsdam). For the separation and quantification of carotenoids (lutein, zeaxanthin, β-cryptoxanthin, canthaxanthin, α- and β-carotene, lycopene) and vitamin A (retinol and retinyl esters), a modified gradient reversed-phase high-performance liquid chromatography (HPLC) system was used. Ethanol (200 µL) were added to 100 µL plasma diluted with 100 µL distilled water (for the correct measurement range) to precipitate proteins. After vortexing for 30 s, plasma was extracted twice with *n*-hexane (1 mL each time stabilized with 0.05% butylated hydroxytoluene) and vortexed for 5 min. Hexane layers were pooled, evaporated under nitrogen, resuspended in 200 µL isopropanol, and 20 µL was injected onto the HPLC-system (Shimadzu, Berlin, Germany). For the separation of the compounds, a C30 carotenoid column, (5 µm, 250 × 3.0 mm; YMC, Wilmington, USA) in line with a C30 pre-column (YMC) with a solvent system consisting of solvent A with methanol/water (90/10; v/v, with 0.4 g/L ammonium acetate in H2O) and solvent B with methanol/methyl-tert-butyl-ether/water (8:90:2; v/v/v, with 0.1 g/L ammonium acetate in H2O) was applied. The flow rate of the mobile phase was 0.2 mL/min and the total run time was 60 min. For peak detection and characterization, a photodiode array detector was used (SPD-M20A, Model 996-Waters, Schimadzu). Carotenoids and retinol were identified by the comparison of the retention time with the external standards and quantified by measuring the absorption at 450 nm for carotenoids and 325 nm for retinol and retinyl esters. The standards for lutein, zeaxanthin, β-cryptoxanthin and lycopene were a kind gift from Hoffmann-La Roche, Basel, Switzerland. The standards for α-carotene, β-carotene, and retinol were purchased from Sigma-Aldrich (Munich, Germany). The limit of detection (LOD), which was defined as analyte amount that produces a signal with a S/N ratio of 3 were 0.0035 µmol/L for retinol, 0.0009 µmol/L for α-carotene, 0.0013 µmol/L for β-carotene, 0.0016 µmol/L for β-cryptoxanthin, 0.0010 µmol/L for lutein, 0.0011 µmol/L for lycopene, and 0.0012 µmol/L for zeaxanthin. The limit of quantification was defined as three times the LOD and were 0.010 µmol/L for retinol, 0.0018 µmol/L for α-carotene, 0.0037 µmol/L for β-carotene, 0.0054 µmol/L for β-cryptoxanthin, 0.0035 µmol/L for lutein, 0.0037 for lycopene, and 0.0035 µmol/L for zeaxanthin. Accuracy and precision were verified using a standard reference material (SMR 968d fat-soluble vitamins in human serum; National Institute of Standards Technology, Gaithersburg, ML, USA) as depicted in Appendix A.

ProVA carotenoid concentrations of each individual were calculated by summing up the concentrations of α-carotene, β-carotene, and β-cryptoxanthin. Vitamin A deficiency was defined as plasma retinol concentrations < 0.7 µmol/L, according to the WHO [1]. The inflammatory marker C-reactive protein (CRP) and the retinol transport proteins retinol-binding protein 4 (RBP4) and transthyretin (TTR), as well as the iron-transport protein ferritin were measured by immunoturbidimetry (Architect 16000, ABBOTT Laboratories, Chicago, USA). Iron deficiency was defined as ferritin < 15 µg/L or as ferritin < 30 µg/L, if CRP was > 0.5 mg/L. Hemoglobin was measured in g/dL using an automated device (Sysmex, Norderstedt, Germany). Participants were considered anemic if hemoglobin concentrations were < 12 g/dL (female) or < 13 g/dL (male) [28].

#### 2.5.2. Genotyping

The DNA has previously been extracted from stabilized full-blood aliquots by using QIAamp DNA blood mini-kit (Qiagen, Hilden, Germany) [13] and was then stored at −20 °C. Using pre-designed TaqMan assays by Life Technologies GmbH (Thermo Fisher Scientific, Darmstadt, Germany) for genotyping, five SNPs were assessed: *BCO1* rs6564851 (C__28949771_10), rs7500996 (C__11513323_10), rs10048138 (C__30021553_10), *PKD1L2* rs6420424 (C__43651006_10) and rs8044334 (C__30706700_10). Analyses were run on the ABI Prism ViiA7 Real-Time PCR system (Applied Biosystems, Forster City, USA) according to the manufacturer’s manual [29,30]. A post-PCR plate read was performed for allelic discrimination using QuantStudio Real-Time PCR System Software version 1.3 by Life Technologies. Water (negative control), prepared reaction mix without DNA (negative control), and duplicate measurements of DNA samples were used as assay controls.

### 2.6. Statistical Analyses

The general characteristics and health status of the study participants are shown across tertiles of total plasma carotenoid concentrations. As the variables were not normally distributed, they are presented as median and interquartile ranges (IQR). For between-group comparisons of continuous variables, Mann–Whitney U tests and Kruskal–Wallis tests were applied for two groups or more groups, respectively. The categorical data are shown as a percentage and number, and an x²-test was used to examine differences between groups. Spearman correlations of plasma carotenoids and retinol concentrations with the FFQ-derived nutrients and intake frequencies of food groups were calculated and are presented as heat maps. Minor allele frequencies of the study population were calculated and compared by visual inspection with the allele frequencies of reference populations in Africa using Ensembl [27]. SNPs were tested for Hardy–Weinberg equilibrium (HWE) [31]. The median and IQR of plasma carotenoids were calculated across the different genotypes for each identified tagging SNP. For this purpose, genotypes were coded as follows:Additive effect:0 = homozygous for normal conversion efficiency;1 = one allele known to decrease conversion efficiency (heterozygous);2 = homozygous for decreased conversion efficiency.Dominant effect:0 = homozygous for normal conversion efficiency;1 = heterozygous or homozygous for decreased conversion efficiency.

To determine the effect on carotenoid concentrations per allele, the additive genotype coding was introduced.

Statistical analyses were performed using the SAS Enterprise Guide version 7.1 (SAS Institute Inc., Cary, NC). A two-tailed *p*-value < 0.05 was considered as statistically significant.

## 3. Results

### 3.1. Socio-Demographic Characteristics

The characteristics of the study population are presented in Table 1. The median age of the 189 adolescents included in the study was 15.2 (15.0–15.3) years, the median BMI was 19.1 (17.6–20.7) kg/m² and sexes were equally represented. The median β-carotene concentration was 1.49 (1.12–2.22) µmol/L with about 36% of the study participants having VAD (plasma retinol: 0.77 (0.64–0.91) µmol/L).

For socio-demographic characteristics, most of the participants lived and attended school in Agogo and were able to read. Adolescents in higher total carotenoid concentration tertiles as compared to the lowest tertiles showed higher illiteracy rates as well as a higher percentage of manually working parents and a lower wealth score. No differences across the tertiles were observed for parental education, number of siblings, and number of people living in the household.

### 3.2. Anthropometric, Dietary and Clinical Factors

Table 2 shows the anthropometric and clinical characteristics of the adolescents related to the varying carotenoid concentrations in the study population. We have observed higher HAZ and BAZ with increasing plasma carotenoids. Further, increasing concentrations of all carotenoids were seen across carotenoid tertiles, with β-carotene as the major component of total carotenoid concentrations, despite similar intakes of RAE. However, the number of vitamin A deficient participants was lower in the highest tertile compared to the lower tertiles of the total plasma carotenoid concentrations, although the plasma retinol concentrations were similar. The study population showed lower plasma retinol concentrations than circulating RBP4. Moreover, hemoglobin concentrations slightly decreased and the percentage of anemic adolescents increased with increasing carotenoid concentrations. Further, no iron deficiency was present in the Ghanaian adolescents.

To further check the impact of nutritional behavior on the varying carotenoid concentrations, Spearman correlations of plasma carotenoid concentrations with calculated FFQ-derived nutrient intakes and daily consumptions of food groups were calculated and illustrated by heat maps. Plasma β-carotene and intake of RAE as well as dietary fiber correlated positively (r = 0.16, *p* = 0.0382; r = 0.15, *p* = 0.0312, respectively) (Figure 1).

Between plasma carotenoid concentrations and the intake of food groups (g/d) (Appendix A), the strongest positive correlations were observed for plasma total carotenoids and α- and β-carotene with plantain (r = 0.27; r = 0.28; r = 0.29, respectively; *p* ≤ 0.0002) and orange/tangerine intake (for all: r = 0.27, *p* = 0.0002). Weaker but also significant positive correlations were found for the intake of cassava, cocoyam and garden egg. No correlations with the dietary intake of animal products were discernible (Appendix A).

### 3.3. Genetic Variants and Plasma Carotenoid Concentrations

The proportions of heterozygotes for each genetic variant ranged between 40–49%. Minor allele frequencies in the population under study for each SNP were 31% (G) for rs6564851, 32% (C) for rs7500996, 38% (A) for rs10048138, 42% (A) for rs6420424, and 43% (T) for rs8044334. All SNPs were in HWE.

The Appendix A present plasma carotenoid and retinol concentrations and the percentages of vitamin A deficient participants across the genotypes, respectively. Carriers of the G allele of SNP rs6564851 had significantly higher proVA carotenoid concentrations (Figure 2).

Accordingly, a significantly higher percentage of G allele carriers was seen among the highest tertile of carotenoid concentration compared to the lowest tertiles (Table 3). The other genetic variants had no impact on plasma carotenoid concentrations. Additionally, retinol concentrations and the number of vitamin A deficient adolescents did not vary across the genotypes (Appendix A).

## 4. Discussion

This study aimed to determine the effects of variants of the BCO1 gene on plasma carotenoid concentrations and the vitamin A status among adolescents in Agogo, rural Ghana. Higher plasma carotenoid concentrations were observed in carriers of the G allele of SNP rs6564851 irrespective of food intake or socioeconomic status, while this variant had no impact on plasma retinol concentrations. No other variant showed associations with carotenoid or retinol concentrations.

### 4.1. Non-Genetic Factors and Carotenoid Concentrations

With regard to sex, our results are in accordance with the findings among European adolescents that boys and girls have similar blood carotenoid concentrations [32]. Clearly, age was not associated with plasma carotenoid concentrations in the present study population because the participants were all born in the same year. The observed higher percentage of manually working parents among adolescents with higher plasma carotenoid concentrations suggests a higher intake of RAE, which was also seen in the FFQ data. Such adequate macro-and micronutrient intakes have favorable effects on the uptake of carotenoids and on the activity of cleaving enzymes [33]. However, in the present study population, neither the intakes of macronutrients nor the biomarkers of iron status explained the discrepancies between carotenoid and retinol concentrations. Nevertheless, plasma retinol concentrations are under homeostatic control and only reflect the vitamin A status of an individual when liver stores are depleted (<0.07 µmol/g) or extremely high (>1.05 µmol/g) [34,35]. Still, liver retinol stores are described to regulate the uptake and cleavage of proVA carotenoids, as observed in Mongolian gerbils [36]. Further, in Filipino schoolchildren the vitamin A status was inversely correlated with the conversion of proVA carotenoids to retinol [37]. To avoid toxic amounts of retinol in the body, the uptake and cleavage of proVA carotenoids is regulated by a negative feedback loop. Retinoic acid activates retinoic acid receptors, which induce the expression of ISX. This inhibits the expression of SR-BI and BCO1, reducing the uptake of proVA and the production of retinol [12].

The sufficient dietary supply with vitamin A is important to maintain the different functions, e.g., vision and cell differentiation, of the body [5,6]. Previous public health efforts have not yielded the desired improvements in vitamin A status in sub-Saharan Africa [2,38]. Food fortification constitutes one of the measures to meet the recommended daily intake. In Ghana, fortified wheat flour and vegetable oil contain added retinyl palmitate to combat VAD. The reasons for the lack of health-beneficial effects might include insufficient concentrations of retinyl palmitate in the fortified foods due to poor storage affecting the stability of this substrate. Given that fortified foods mainly reach customers in urban areas [39,40], alternative approaches to ameliorate VAD in sub-Saharan Africa comprise the promotion of dietary diversification and increased intakes of vitamin A-rich foods.

Moreover, VAD is likely to be overestimated during inflammation (CRP > 5 mg/L). RBP4 is a transport protein of retinol but also a negative acute-phase protein which is downregulated in the inflammatory state. Then, lower concentrations of plasma retinol can be seen [41]. Our study participants showed normal mean RBP4 and mean CRP concentrations. For retinol concentrations, VAD cut-offs are independent of the inflammatory state because hepatic retinol stores remain adequate even when plasma retinol decreases during inflammation. However, higher RBP4 than retinol concentrations were observed in the Ghanaian adolescents, which cannot be explained in detail. Retinol and RBP4 was not always present as 1:1 ratio in the human plasma [42]. Elevated levels of RBP4 are observed in patients with renal dysfunction or in obese subjects and both conditions can be excluded as causing factors in our cohort [43,44]. Moreover, we cannot exclude the presence of SNPs in the RBP4 gene [19]. According to Zanotti and Berni, healthy individuals exhibit a RBP4:TTR ratio of 0.4 [45], which was seen among the adolescents in the present study. Hence, the synthesis and the secretion of these retinol transport proteins appears to be unaffected.

### 4.2. BCO1 Variants and Carotenoid Concentrations

To date, only a few studies have described the same phenomenon that we observed in the Agogo 2000 birth cohort study [46,47]. Genetic variants have previously contributed to the interindividual variability of plasma carotenoids, mainly in non-African populations [20]. West et al. have reviewed the data from low-income countries with respect to the conversion efficiency of proVA to retinol. They have reported lower conversion rates as compared to populations living in high-income countries (means: 21 µg vs. 12 µg of β-carotene converted into 1 µg of RAE) [48]. Consequently, the intake of vitamin A precursors needs to be higher in low-income countries to meet the same adequate vitamin A supply as in high-income countries. However, the reasons for these observations have not yet been clarified. Variations in gene encoding for enzymes responsible for the absorption, cleavage and distribution of vitamin A can modulate the effect of diet on this micronutrient status [20]. These gene–diet interactions possibly explain the observed interindividual differences in plasma carotenoid concentrations.

The cleaving enzyme BCO1 constitutes one of the limiting factors for the conversion of proVA carotenoids; SNPs in this gene have been described in this regard by different authors [17,20,22,49]. Most reports describe the SNP rs6564851 in this gene, which was also significantly associated with higher carotenoid concentrations in our study population. Lobo et al. identified that this variant impacts on vitamin A production through the regulation of ISX expression. Carriers of this variant have shown up to 50% higher fasting blood β-carotene concentrations than non-carriers [49]. However, most studies describing the effect of BCO1 gene variant rs6564851 were conducted in adults and non-African populations [18,22,50]. In BCO1 knockout mice that were fed a β-carotene-rich diet, the lack of BCO1 and ISX led to an accumulation of β-carotene and increased serum lipids [51]. Up to now, blood lipids have not been measured in the present study population. Other *BCO1* SNPs under study here might have less impact on carotenoid conversion and thus, we may need larger samples sizes to detect their effects.

### 4.3. Strengths and Limitations

The present analysis contributes to the small number of studies analyzing the interindividual variability of plasma carotenoid concentrations in sub-Saharan African populations. Given that our sample size was moderate, only genetic variants with a minor allele frequency of ≥ 30% were chosen for this analysis. Thus, we might have excluded other important variants in the vitamin A metabolism that possibly explain varying carotenoid concentrations. BCO1 variant rs6564851 cannot exclusively be associated with alterations of plasma proVA concentrations. In addition, genetic variants involved in proVA absorption, metabolism and distribution might affect carotenoid concentrations among adolescents in rural Ghana. Further, the energy and nutrient intakes calculated from FFQ data serve to rank our study participants by their consumption, while absolute intakes cannot be estimated by this dietary assessment method. For more accurate calculation of dietary intakes, repeated 24-h dietary recalls or vitamin A-specific screeners will be necessary, and the refinement of FFQ portion sizes for children and adolescents is desirable. Finally, we cannot comment on the influence of blood lipoproteins and biomarkers of zinc concentration. Still, the distribution of genetic variants in our study population are random and independent from dietary intakes, according to Mendel’s rules of inheritance.

## 5. Conclusions

To date, food fortification is established in affected countries to improve the vitamin A status of populations at risk. Regarding the presence of genetic variant encoding in the cleaving enzyme BCO1, preformed vitamin A should be considered as the main component to be added to foods. Further, introducing biofortification of plants high in β-carotene might have a reduced benefit on retinol concentrations in the body when the conversion efficiency of the BCO1 gene product is impaired. Therefore, it is warranted to investigate the role of BCO1 variants for the success of supplementation programs and fortification efforts among vulnerable populations in sub-Saharan Africa.

## Figures and Tables

**Figure 1 nutrients-12-01786-f001:**
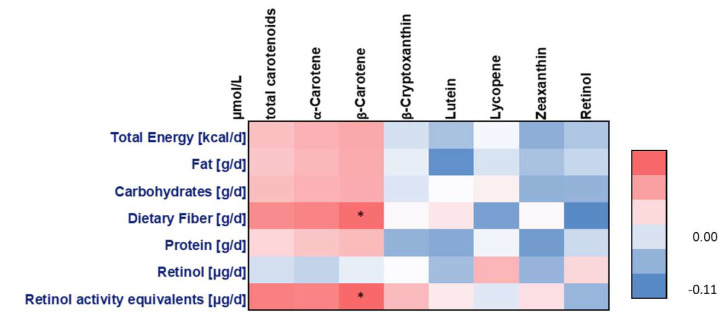
Spearman correlations of calculated food frequency questionnaire-derived nutrient intakes with plasma carotenoid and retinol concentrations; * *p* < 0.05.

**Figure 2 nutrients-12-01786-f002:**
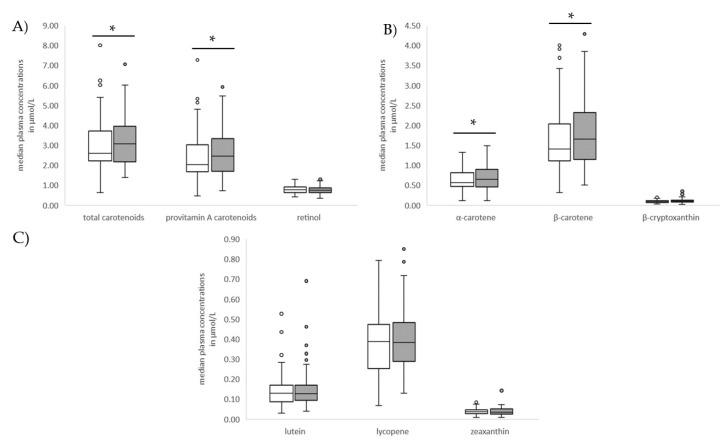
Median plasma: (**A**) total carotenoid, provitamin A carotenoid, and retinol; (**B**) provitamin A carotenoid (α-carotene, β-carotene, and β-cryptoxanthin); (**C**) non-provitamin A carotenoid (lutein, lycopene, and zeaxanthin) concentrations across the genotypes in single nucleotide polymorphism (SNP) rs6564851.TT (white box, *n* = 89); GT/GG (grey box, *n* = 100). Data presented as medians and interquartile ranges and were compared by Mann–Whitney U test, * *p* < 0.05

**Table 1 nutrients-12-01786-t001:** Socio-demographic characteristics of the study participants across tertiles of plasma total carotenoid concentration.

	Total Carotenoid Concentration [µmol/L]			
	Total (*n* = 189) 2.81 (2.17–3.78)	First Tertile (*n* = 63) 1.97 (1.61–2.17)	Second Tertile (*n* = 63) 2.81 (2.55–3.09)	Third Tertile (*n* = 63) 4.19 (3.78–4.90)
**Sex (m/f)**	94/95	32/31	31/32	31/32
**Age (in years)**	15.2 (15.0–15.3)	15.2 (15.1–15.3)	15.2 (15.0–15.3)	15.2 (15.0–15.3)
**Residence, Agogo %**	71.4 (135)	68.3 (43)	74.6 (47)	71.4 (45)
**Place of school, Agogo %**	57.1 (108)	55.6 (35)	54.0 (34)	61.9 (39)
**Illiteracy %**	9.0 (17)	6.4 (4)	6.4 (4)	14.3 (9)
**Education %**				
**Mother**				
None	4.2 (8)	4.8 (3)	0 (0)	7.9 (5)
Primary	18.0 (34)	20.6 (13)	20.6 (13)	12.7 (8)
Secondary	18.5 (35)	25.4 (16)	12.7 (8)	17.5 (11)
Tertiary	3.2 (6)	4.8 (3)	1.6 (1)	3.2 (2)
Other	56.1 (106)	44.4 (28)	65.1 (41)	58.7 (37)
**Father**				
None	3.7 (7)	3.2 (2)	4.8 (3)	3.2 (2)
Primary	12.2 (23)	15.9 (10)	14.3 (9)	6.3 (4)
Secondary	29.1 (55)	33.3 (21)	23.8 (15)	30.2 (19)
Tertiary	8.5 (16)	6.3 (4)	11.1 (7)	7.9 (5)
Other	46.5 (88)	41.3 (26)	46.0 (29)	52.4 (33)
**Occupation %**				
**Mother**				
Manual work	22.7 (43)	19.1 (12)	19.1 (12)	30.2 (19)
Intellectual work	63.5 (120)	68.2 (43)	69.8 (44)	52.4 (33)
Other work	10.6 (20)	11.1 (7)	9.5 (6)	11.1 (7)
Unemployed	3.2 (6)	1.6 (1)	1.6 (1)	6.3 (4)
**Father**				
Manual work	43.4 (82)	39.8 (25)	42.8 (27)	47.6 (30)
Intellectual work	30.2 (57)	39.6 (25)	27.0 (17)	23.8 (15)
Other work	25.9 (49)	20.6 (13)	28.6 (18)	28.6 (18)
Unemployed	0.5 (1)	0 (0)	1.6 (1)	0 (0)
**Number of siblings**	4 (3–5)	4 (3–5)	4 (3–5)	4 (3–5)
**People in the household**	11 (7–20)	10 (8–20)	11 (7–20)	12 (7–18)
**Wealth score**	0.55 (0.36–0.64)	0.55 (0.36–0.64)	0.55 (0.36–0.73)	0.46 (0.27–0.64)

Continuous variables are shown as medians (interquartile ranges); categorical variables are presented as percentages (number); m: male, f: female; wealth score was calculated as the number of existing assets per the 11 possible items.

**Table 2 nutrients-12-01786-t002:** Anthropometric and clinical characteristics across tertiles of plasma total carotenoid concentration.

	Total Carotenoid Concentration [µmol/L]			
	Total (*n* = 189) 2.81 (2.17–3.78)	First Tertile (*n* = 63) 1.97 (1.61–2.17)	Second Tertile (*n* = 63) 2.81 (2.55–3.09)	Third Tertile (*n* = 63) 4.19 (3.78–4.90)
BMI [kg/m²]	19.1 (17.6–20.7)	19.4 (17.9–21.7)	19.0 (17.7–20.8)	19.0 (17.5–20.2)
HAZ	−0.87 (−1.55–0.28)	−0.70 (−1.47–0.03)	−0.83 (−1.67–0.24)	−1.17 (−1.57–0.52)
BAZ	−0.40 (−1.07–0.19)	−0.32 (−0.96–0.59)	−0.39 (−1.01–0.13)	−0.54 (−1.11–0.02)
Energy intake [kcal/d]	2872 (2424–3440)	2748 (2380–3436)	2921 (2445–3440)	2961 (2424–3490)
CHO [energy%]	48.2 (44.3–51.5)	47.2 (44.7–50.7)	49.0 (44.2–52.4)	48.0 (44.2–51.2)
Fat [energy%]	37.1 (34.6–40.2)	37.7 (35.4–40.2)	36.6 (34.1–40.1)	37.9 (34.3–41.3)
Protein [energy%]	14.5 (13.5–15.4)	14.5 (13.7–15.5)	14.6 (13.4–15.4)	14.2 (13.5–15.3)
RAE [µg/d]	1523 (1229–1879)	1440 (1109–1879)	1545 (1214–1825)	1529 (1315–2015)
Physical activity [min/day] ^a^	179 (167–196)	179 (168–198)	181 (168–195)	177 (164–196)
Blood pressure [mmHg]				
Systolic	110 (103–117)	110 (102–117)	109 (102–118)	112 (104–119)
Diastolic	68 (62–74)	68 (63–73)	66 (61–74)	70 (64–75)
**Blood parameters**				
α-Carotene [µmol/L]	0.61 (0.46–0.86)	0.39 (0.32–0.47)	0.60 (0.54–0.66)	0.94 (0.86–1.10)
β-Carotene [µmol/L]	1.49 (1.17–2.22)	0.97 (0.79–1.12)	1.49 (1.37–1.77)	2.52 (2.22–2.96)
β-Cryptoxanthin [µmol/L]	0.09 (0.07–0.12)	0.06 (0.04–0.92)	0.10 (0.08–0.13)	0.11 (0.09–0.15)
Lutein [µmol/L]	0.13 (0.09–0.17)	0.08 (0.07–0.12)	0.13 (0.10–0.18)	0.16 (0.13–0.20)
Lycopene [µmol/L]	0.38 (0.28–0.48)	0.33 (0.22–0.43)	0.39 (0.30–0.50)	0.42 (0.30–0.59)
Zeaxanthin [µmol/L]	0.04 (0.03–0.05)	0.03 (0.02–0.04)	0.04 (0.03–0.06)	0.05 (0.03–0.06)
Retinol [µmol/L]	0.77 (0.64–0.91)	0.79 (0.61–0.92)	0.75 (0.62–0.89)	0.75 (0.66–0.93)
VAD %	35.5 (67)	36.5 (23)	39.7 (25)	30.2 (19)
RBP4 [µmol/L] ^b^	1.17 (0.91–1.42)	1.22 (0.89–1.53)	1.18 (0.92–1.42)	1.16 (0.92–1.34)
TTR [µmol/L] ^b^	2.79 (2.25–3.51)	2.93 (2.27–3.55)	2.66 (2.20–3.31)	2.91 (2.33–3.53)
RBP4/TTR ^b^	0.41 (0.33–0.50)	0.41 (0.36–0.47)	0.40 (0.32–0.53)	0.42 (0.33–0.49)
CRP [mg/L] ^b^	0.62 (0.30–2.11)	0.57 (0.29–2.16)	0.69 (0.42–2.18)	0.56 (0.10–1.34)
Hemoglobin [g/dL]	13.0 (12.2–14.0)	13.2 (12.1–14.1)	13.0 (12.4–13.9)	12.9 (12.1–14.0)
Anemia (positive %)	30.7 (58)	30.2 (19)	27.0 (17)	34.9 (22)
Ferritin [µg/L] ^b^	57.8 (39.9–90.0)	62.4 (40.1–90.0)	50.7 (39.1–77.9)	64.2 (40.6–96.4)
Fasting plasma glucose [mmol/L]	4.3 (3.9–4.7)	4.2 (3.8–4.6)	4.3 (3.9–4.8)	4.3 (3.9–4.8)

Continuous variables are presented as median (interquartile range); categorical variables shown as percentage (number); BAZ: BMI-for-age z-score; BMI: body mass index; CHO: carbohydrates; CRP: C-reactive protein; HAZ: height-for-age z-score; RAE: retinol activity equivalents; RBP4: retinol-binding protein 4; TTR: transthyretin; VAD: vitamin A deficiency (retinol < 0.7 µmol/L); anemia: hemoglobin < 12 g/dL (male), hemoglobin < 13 g/dL (female). ^a^ n = 187 (62/62/63) ^b^ n = 180 (61/60/59).

**Table 3 nutrients-12-01786-t003:** Distribution of the alleles known to decrease the conversion efficiency of β-carotene 15,15’-oxygenase (BCO1) across tertiles of plasma total carotenoid concentration [µmol/L].

	First Tertile (*n* = 63) 1.97 (1.61–2.17)	Second Tertile (*n* = 63) 2.81 (2.55–3.09)	Third Tertile (*n* = 63) 4.19 (3.78–4.90)	x² *p*-Value
**rs6564851** **GT/GG**	42.9 (27)	50.8 (32)	65.1 (41)	0.041
**rs7500996** **CT/CC**	50.8 (32)	52.4 (33)	54.0 (34)	0.938
**rs10048138** **AG/AA**	58.7 (37)	69.8 (44)	58.7 (37)	0.331
**rs6420424** **AG/AA**	69.8 (44)	63.5 (40)	66.7 (42)	0.752
**rs8044334** **GT/GG**	82.5 (52)	79.4 (50)	82.5 (52)	0.869

Categorical variables are shown as percentages (numbers).

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
