# Peer review of "SNP rs6564851 in the BCO1 Gene Is Associated with Varying Provitamin a Plasma Concentrations but Not with Retinol Concentrations among Adolescents from Rural Ghana"

_nutrients, 2020, doi:10.3390/nu12061786_

Round 1
Reviewer 1 Report
While it is now well accepted that certain SNPs within and upstream of the BCO1 enzyme impact the cleavage of provitamin A carotenoids into vitamin A, the role these functional polymorphisms play in impacting vitamin A status in historically deficient groups is not well appreciated. The current study attempts to determine the role of these SNPs in vitamin A status of rural Ghanan adolescents (n = 189) - a region still plagued with vitamin A deficiency despite fortification efforts.
Major concerns:
The authors claim that their results suggest, “introducing biofortification of plants high in β-carotene might have reduced benefit on retinol concentrations of the body, when the conversion efficiency of the BCO1 gene product is impaired.” However, they have no grounds to make this assessment based on the data collected from the subjects. They failed to assess the “true” vitamin A status of the study subjects (typically performed using a retinol isotope tracer measure to determine liver stores OR a liver biopsy), and do not seem to recognize/appreciate that blood retinol levels alone are a poor measure of status. Based on the RBP to TTR ratio (which needs to be reported in “Results” and not just mentioned in passing in “Discussion”), these were well nourished and infection-free adolescents who were not suffering from deficiency to begin with. The authors would do better to position this work as testing the effects of these SNPs in a distinct population that has not previously been tested.
Additional issues:
- Provide RBP to TTR ratio in Table 2
- Methods section 2.2 – Were children assessed for infection? Was being healthy an inclusion criteria and acute or chronic illness an exclusion criteria?
- Methods section 2.5 – was blood protected from sunlight (i.e. coming from a window) during collection? Was it processed (i.e. centrifuged and flash frozen) immediately after collection to prevent loss of carotenoids and retinol to oxidation…or did some samples sit on the lab bench for 4 hours before the plasma was separated from the H&H. This can dramatically impact both carotenoid and vitamin A levels
- Methods section 2.5.1 – No detail is given on the sample extraction nor the analysis.
- The authors imply but do not directly state that the method in reference 22 was used – they need to clarify if the method was extraction, analysis, or both. Additionally, some basic info needs to be added to this manuscript (even if ref. 22 was followed precisely)
- Which HPLC column was used, and what solvents were in the gradient?
- High performance liquid chromatography separates carotenoids in a sample, but it does not “detect” them – note which platform was used for detection (i.e. DAD, MS)?
- What was the injection volume of each sample?
- Please provide the LOD and LOQ estimated for the method of detection. Assessing these compounds in 100 uL of plasma is often below mass spec. and DAD LOQ for prospective feeding studies where levels are typically substantially higher.
- Were external calibration curves used for quantitation?
- Was an internal standard used? If so, which one, and how was it used (i.e. to correct for recoveries or instrument response or both)?
- Figure 2 – provide median and SEM’s – showing 25th and 75th percentiles as a bar chart is confusing and not particularly informative. The median is a better graphical choice if the data is non-normally distributed OR a box-and-whisker plot.
- Table S1 – “vitamin A deficient” column, provide units with the column header
- Weird spacing throughout the manuscript, double check text for formatting issues.
Author Response
Please, see attached document.

Reviewer 2 Report
Title: SNP rs6564851 in the BCO1 gene is associated with varying pro-vitamin A plasma concentrations but not with retinol concentrations among adolescents from rural Ghana
Manuscript ID: nutrients-792689
Description:
Vitamin A is an essential nutrient required for embryonic development, ocular function, and immune function. Vitamin A deficiency (VAD) is associated with malnutrition, infection, and night blindness. According to the World Health Organization in 2013, VAD was and continues to be a public health problem affecting one third of children aged 6 to 59 months. Vitamin A fortification in food can reduce the VAD in many sub-Saharan African countries. Several genetic prevalence studies are needed to explain VAD based on ethnicity. The present study represents the single nucleotide polymorphism in the Vitamin A forming enzyme, BCO1, which results in the alteration of pro-vitamin A concentration, but not retinol concentration in rural Ghana adolescents. Listed below are some comments that need to be addressed in this manuscript.
Comments
- Introduction: The manuscript is focus on SNP of BCO1 gene and therefore, it is important to include the previous reports about BCO1 genetic variants in the introduction section.
- Author has selected only six BCO1 variants with excluding the allele frequency < 30% but authors has not included the BCO1 variant rs12934922 in the present study which has 40% of allele frequency.
- Table -2 shows that the retinol and RBP4 levels are different; however, retinol and RBP4 are present in 1:1 ratio in serum. As a result, the author needs to address this discrepancy.
- Vitamin A can passively diffuse through intestine, but pro-vitamin A needs SR-B1 for intestinal absorption into the epithelial cells. Thus, it is important to determine any variants influencing serum pro-vitamin A concentration in the adolescent of rural Ghana.
- The author has pointed out that there is no difference in retinol concentration observed between genotypes. The serum retinol levels has no relationship with serum pro-vitamin A level because it is homeostatic. Pro-vitamin A can differ based on dietary intake and hepatic storage can be sufficient to maintain vitamin A levels. Author needs to address this discrepancy.
- The author has not discussed similar kind of research published about SNP in BCO1 affecting serum pro-vitamin A in this manuscript. Therefore, reviewer suggests discussing about the similar kind of research in discussion, which will help to increase the scope of this research.
- The author has pointed that due to small number of sample size they excluded the other BCO1 variants that might lead to varying carotenoid concentration. Without analyzing all major BCO1 variants, it is difficult to claim that the BCO1 variant rs6564851 is associated with alterations of pro-vitamin A in serum.
- Vitamin A and pro-vitamin can be obtained through dietary source only. Thus, the need to perform a more reliable dietary intake assessment method from subjects is needed in the present study.
Author Response
Please, see the attachment.

Round 2
Reviewer 1 Report
The authors did a very nice job of addressing previous concerns. However, I think it's important to add a line about the primary aim of the parent study in L104-105 (i.e. to assess the relationships between gestational malaria and health outcomes), and that this was a secondary analysis of the samples.
Author Response
Thank you for addressing this point. We have now clarified in lines 104-105 that "the present work constitutes a secondary analysis."
Reviewer 2 Report
Title: SNP rs6564851 in the BCO1 gene is associated with varying pro-vitamin A plasma concentration but not with retinol concentrations among adolescents from rural Ghana
Manuscript ID: nutrients-792689
Comments
- Author has selected only BCO1 variants with excluding the allele frequency < 30% but authors has not included the BCO1 variant rs12934922 in the present study and the reviewer raises a concern regarding this and the author has not addressed the comments clearly.
- HPLC trace for retinol, Pro vitamin A measured from serum samples can be included in this manuscript this will increase the scope of this manuscript.
- Line no (395-398) this description is more shallow need to be more detailed and also without analyzing the variations in the gene involved in provitamin A absorption and metabolism one cannot conclude variation in pro-vitamin A plasma concentration by SNP in pro vitamin A cleaving enzyme BCO1.
- The serum RBP4 and retinol levels are inappropriate in the current study, Because RBP4 is the major marker to measure retinol level in serum. The RBP4 and retinol was present in 1:1 ratio in healthy individuals and it will only violate in individuals with kidney problem and obesity (Mills et al 2003, Gibson R, ed. Principles of nutritional assessment. Oxford, UK, Oxford University Press, 2005, Biochemistry of Vitamin A by Jagannath Ganguly). So it is unclear in the present study why there is a variation in serum retinol and RBP4 level.
- The dietary intake assessment of this current study is not more reliable and only by this assessment method it is difficult to come to a strong conclusion “Revised prevention methods are necessary in countries where, to date, no improvement of the vitamin A status is discernible and the access to fortified products needs to be improved for people living in rural areas”
Author Response
- Author has selected only BCO1 variants with excluding the allele frequency < 30% but authors has not included the BCO1 variant rs12934922 in the present study and the reviewer raises a concern regarding this and the author has not addressed the comments clearly.
Response: We kindly refer the reviewer to our previous explanation in the rebuttal letter, dated 18th May 2020: We have been alarmed by this comment and thus, have carefully checked our SNP selection using the latest literature. In fact, the variant rs12934922 reaches MAF >30% in non-African populations. While this is not the case for our sub-Saharan African population (http://www.ensembl.org/Homo_sapiens/Variation/Population?db=core;r=16:81267589-81268589;v=rs12934922;vdb=variation;vf=24787405), we did not test this SNP in the present analysis.
We now highlight this fact in the Methods section (line 154-155): “Due to the small study population, SNPs with an allele frequency of <30% in African reference populations (Ensembl, version 95 [27]) (n=8) and variants in LD (D’≥0.8; r²>0.5) (n=2) were excluded, […]”
- HPLC trace for retinol, Pro vitamin A measured from serum samples can be included in this manuscript this will increase the scope of this manuscript.
Response: Thank you. We also find this important. Therefore, the concentrations of the main provitamin A carotenoids alpha-carotene, beta-carotene and beta-cryptoxanthin are provided in Table 2.
- Line no (395-398) this description is more shallow and need to be more detailed and also without analyzing the variations in the gene involved in provitamin A absorption and metabolism one cannot conclude variation in pro-vitamin A plasma concentration by SNP in pro vitamin A cleaving enzyme BCO1.
Response: Thank you. We now clearly state in lines 397-399: “In addition, genetic variants involved in proVA absorption, metabolism and distribution might affect carotenoid concentrations among adolescents in rural Ghana.”
- The serum RBP4 and retinol levels are inappropriate in the current study, Because RBP4 is the major marker to measure retinol level in serum. The RBP4 and retinol was present in 1:1 ratio in healthy individuals and it will only violate in individuals with kidney problem and obesity (Mills et al 2003, Gibson R, ed. Principles of nutritional assessment. Oxford, UK, Oxford University Press, 2005, Biochemistry of Vitamin A by Jagannath Ganguly). So it is unclear in the present study why there is a variation in serum retinol and RBP4 level.
Response: It is true, that retinol and RBP concentrations are not present in a 1:1 ratio in the current study, but we cannot understand why the serum RBP4 and retinol levels should be inappropriate. The methods we used for analyses are established and were published several times e.g. (Alicke et al, 2017; Danquah et al, 2015; Longardt et al, 2014). We fully agree that plasma retinol or RBP4 concentrations are only one of several biomarkers for assessing vitamin A status (Tanumihardjo et al, 2016). Theoretically, serum RBP4 correlates closely with serum retinol in subjects with normal liver and kidney function who are not obese. In the present study we found a molar excess of RBP4 over retinol meaning that all retinol is saturated with RBP4 and a proportion of RBP4 is unbound to retinol (as apo-RBP4). This is not unexpected, because apo-RBP4 originates after the delivery of retinol to peripheral tissue and the subsequent disruption of the RBP4-TTR complex. Many authors have described this observation. For instance, Smith and Goodman (1971) reported in an early study molar values for RBP to vitamin A of 1.22 ± 0.04 in normal subject (Smith et al, 1971). Mills et al. (2008) found a molar excess of RBP4 over retinol in nonobese adults. Sankaranarayanan et al. (2005) also found higher molar RBP4 than retinol. Indeed, we cannot explain the causes in detail, but we are sure that the data are reliable and may provide a good basis for further investigations.
We addressed this in lines 358-362: “However, higher RBP4 than retinol concentrations could be observed in the Ghanaian adolescents, which cannot be explained in detail. Retinol and RBP4 was not always are present 1:1 in the human plasma [42]. Elevated levels of RBP4 are observed in patients with renal dysfunction or in obese subjects and both conditions can be excluded as causing factors in our cohort [43,44].“
However, the RBP:TTR ratio shows no abnormalities, and the adolescents can be considered healthy (Zanotti et al, 2004).
- The dietary intake assessment of this current study is not more reliable and only by this assessment method it is difficult to come to a strong conclusion “Revised prevention methods are necessary in countries where, to date, no improvement of the vitamin A status is discernible and the access to fortified products needs to be improved for people living in rural areas”
Response: Indeed, it is a strong conclusion. Thus, we have mitigated our statement (lines 414-415): “Therefore, it is warranted to investigate the role of BCO1 variants for the success of supplementation programs and fortification efforts among vulnerable populations in SSA.”
References:
Alicke, M.; Boakye-Appiah, J.K.; Abdul-Jalil, I.; Henze, A.; van der Giet, M.; Schulze, M.B.; Schweigert, F.J.; Mockenhaupt, F.P.; Bedu-Addo, G.; Danquah, I. Adolescent health in rural Ghana: A cross-sectional study on the co-occurrence of infectious diseases, malnutrition and cardio-metabolic risk factors. PLoS One 2017, 12, e0180436.
Danquah, I.; Dobrucky, C.L.; Frank, L.K.; Henze, A.; Amoako, Y.A.; Bedu-Addo, G.; Raila, J.; Schulze, M.B.; Mockenhaupt, F.P.; Schweigert, F.J. Vitamin A: potential misclassification of vitamin A status among patients with type 2 diabetes and hypertension in urban Ghana. Am. J. Clin. Nutr. 2015, 102, 207–214.
Longardt, A.C.; Schmiedchen, B.; Raila, J.; Schweigert, F.J.; Obladen, M.; Bührer, C.; Loui, A. Characterization of the vitamin A transport in preterm infants after repeated high-dose vitamin A injections. Eur. J. Clin. Nutr. 2014, 68, 1300–1304.
Mills, J.P.; Furr, H.C.; Tanumihardjo, S.A. Retinol to retinol-binding protein (RBP) is low in obese adults due to elevated apo-RBP. Exp. Biol. Med. (Maywood). 2008, 233, 1255–1261.
Sankaranarayanan, S.; Suárez, M.; Taren, D.; Genaro-Wolf, D.; Duncan, B.; Shrestha, K.; Shrestha, N.; Rosales, F.J. The concentration of free holo-retinol binding protein is higher in vitamin A-sufficient than in deficient Nepalese women in late pregnancy. J. Nutr. 2005, 135, 2817–2822.
Smith, F.R.; Goodman, D.S. The effects of diseases of the liver, thyroid, and kidneys on the transport of vitamin A in human plasma. J. Clin. Invest. 1971, 50, 2426–2436.
Tanumihardjo, S.A.; Russell, R.M.; Stephensen, C.B.; Gannon, B.M.; Craft, N.E.; Haskell, M.J.; Lietz, G.; Schulze, K.; Raiten, D.J. Biomarkers of Nutrition for Development (BOND)-Vitamin A Review. J Nutr 2016, 146, 1816S–48S.
Zanotti, G.; Berni, R. Plasma retinol-binding protein: structure and interactions with retinol, retinoids, and transthyretin. Vitam Horm 2004, 69, 271–295.